# Q-STRONG: Quantum-Statistical Robustness with Noise-Guarded Dynamics for Learning Conference Submissions

## Abstract

State-of-the-art learners remain fragile under heavy-tailed noise, adversarial perturbations and decoherence. We propose *Q-STRONG*, a quantum–statistical framework for certified robust learning that uses the spectral structure of a learned state representation as a stability signal. Inputs are embedded into a normalized quantum state space, and a task-aligned Hamiltonian induces a low-energy representation whose spectral gap $\Delta_\theta(x)$ quantifies local stability. This gap steers both training and certification: during optimization, robust losses and quantile-based clipping reduce gradient tail effects; at inference, a gap-adaptive randomized smoothing scheme chooses the noise level $\sigma(x) = \kappa \Delta_\theta(x)^{-\beta}$, yielding larger certified $\ell_2$ radii exactly where the representation is stable. We provide non-asymptotic guarantees for quantile-clipped robust SGD, stability-based generalization bounds with improved effective smoothness, and gap-adaptive extensions of randomized-smoothing certificates tied to $\Delta_\theta(x)$. Empirically, Q-STRONG attains a favorable accuracy–robustness frontier on MNIST and CIFAR-10 under label noise and common corruptions, and on synthetic manifolds that stress intrinsic dimension and outliers, while adding modest overhead and thus offers a practical, theoretically grounded route to certified, noise-resilient learning.

## 1 Introduction

Recent progress in deep learning has highlighted a persistent challenge: despite excelling on clean benchmarks, modern models remain highly sensitive to small perturbations or structural noise in the input distribution and remain brittle under heavy–tailed corruptions, label noise, covariate shift, and adversarial perturbations (Goodfellow et al., 2015; Madry et al., 2018). , mislabeled examples, and gradient-instability during training can severely distort the learned representation. Meanwhile, methods for certified robustness—such as randomized smoothing—provide formal guarantees but operate agnostically to the internal geometry of the model and often yield conservative bounds. At the same time, quantum-inspired representations have shown that spectral properties of state embeddings, such as local energy gaps, correlate strongly with the intrinsic stability of features and their resistance to perturbations. These observations motivate the need for a unified framework that leverages stability information from the representation itself while still remaining compatible with classical hardware and learning pipelines.

These vulnerabilities are further amplified in stochastic or resource–constrained regimes—e.g., near–term quantum processors (NISQ) where readout noise, crosstalk, and decoherence are intrinsic (Preskill, 2018). Building *robust* and *resilient* systems therefore requires joint progress on (i) *statistical* objectives whose influence functions temper outliers, (ii) *optimization* procedures that suppress instability from rare, large gradients, and (iii) *certification* methods that turn empirical robustness into verifiable guarantees. On emerging noisy intermediate-scale quantum (NISQ) hardware, an additional layer of stochasticity arises from gate errors, decoherence, and readout noise (Preskill, 2018) , further stressing robustness and reliability. Designing learners that remain accurate, stable to training noise, and *certifiably* robust at test time is therefore a central challenge.

We introduce Q-STRONG (*Quantum–Statistical Robustness with Noise–Guarded Dynamics for Learning*), a quantum–statistical framework that unifies robust M–estimation, quantile–scheduled

gradient clipping, and adaptive randomized smoothing within a principled state–space formulation. Classical inputs are embedded as quantum states by a trainable encoder; a task–aligned Hamiltonian induces a low–energy representation whose *spectral gap* serves as a stability indicator. During training we minimize a robust loss (e.g., Huber, Catoni) to bound per–sample influence (Huber, 1964; Catoni, 2012), while applying *dynamic clipping* that sets the clipping norm to a running quantile of per–sample gradient norms, suppressing rare but destabilizing updates (Menon et al., 2020; Ye et al., 2025). At inference, we deploy *noise–guarded randomized smoothing*: Gaussian perturbations are injected with variance $\sigma(x) \propto 1/\Delta(x)$, where $\Delta(x)$ is the empirical spectral gap of the learned quantum representation. This gap–adaptive schedule enlarges certified $\ell_2$ radii when the representation is stable, linking certification to state–space dynamics (Cohen et al., 2019; Salman et al., 2019; Lyu et al., 2024).

Robust statistics offers estimators with bounded influence functions that curb heavy–tailed noise and contamination (Huber, 1964; Catoni, 2012). Randomized smoothing converts any base classifier into a certifiable one by majority vote under Gaussian noise, yielding instance–wise robustness radii (Cohen et al., 2019; Salman et al., 2019; Yang et al., 2020). Gradient clipping is a pragmatic stabilizer, but naïve clipping alone is not label–noise robust; partially Huberised/composite loss strategies and *optimized* schedules address this limitation (Menon et al., 2020; Ye et al., 2025). In parallel, quantum machine learning (QML) connects quantum embeddings to kernel methods (Schuld & Killoran, 2019), has strong representational promise (Biamonte et al., 2017), but faces NISQ realities and barren plateau phenomena (Preskill, 2018; McClean et al., 2018). Recent QML efforts show noise–aware representation/observable learning and provably noise–resilient training (Candelori et al., 2024; Khanal & Rivas, 2024; Tecot et al., 2025). *Q-STRONG* bridges these threads: robust objectives and dynamic clipping are enforced in the quantum–embedded space, and certification is made *gap–adaptive*—tying guarantees to physically meaningful stability signals. Non–asymptotic analysis for weakly smooth robust objectives: (i) convergence of clipped SGD to stationary points with constants controlled by the clipping quantile; (ii) a stability–based generalization bound in which the effective Lipschitz constant is reduced by robustification and clipping; (iii) transfer of smoothing certificates (Cohen et al., 2019; Salman et al., 2019) to a quantum readout with *gap–adaptive noise*, yielding larger radii on stable representations; and (iv) parameter–noise resilience bounds that tie prediction drift under hardware perturbations to inverse powers of the spectral gap.

## 2    RELATED WORK

### 2.1    ROBUST STATISTICS AND ROBUSTIFICATION

Classical robust methods (Huber, redescending/Catoni) bound the influence of outliers and stabilize estimation in heavy–tailed regimes (Huber, 1964; Catoni, 2012). These techniques extend to modern ML as robust losses and reweighting schemes, but by themselves do not provide adversarial guarantees or optimization stability under rare gradient spikes.

### 2.2    GRADIENT CLIPPING AND NOISE-AWARE OPTIMIZATION

Gradient clipping is widely used to avoid exploding updates, yet standard clipping alone is not label–noise robust; its effect is equivalent to a fully Huberised loss that remains vulnerable under symmetric noise (Menon et al., 2020). Composite/partially Huberised losses improve robustness (Menon et al., 2020), and *optimized* clipping schedules that adapt thresholds over training further enhance performance under label noise (Ye et al., 2025).

### 2.3    CERTIFIED ROBUSTNESS VIA RANDOMIZED SMOOTHING

Randomized smoothing scales certification to large models by turning any base classifier into a smoothed classifier with instance-wise $\ell_2$ radii (Cohen et al., 2019). Adversarially trained smoothing improves the accuracy–certificate frontier (Salman et al., 2019). Extensions broaden the noise families and certification theory (Yang et al., 2020; Mohapatra et al., 2020), and recent adaptive variants certify multi-step/test-time adaptation (Lyu et al., 2024). Our work contributes a *gap–adaptive* smoothing schedule that ties $\sigma(x)$ to quantum stability.

## 2.4 QUANTUM MACHINE LEARNING UNDER NOISE

QML promises expressive embeddings and kernel–like advantages (Biamonte et al., 2017; Schuld & Killoran, 2019) but faces NISQ noise and barren plateaus (Preskill, 2018; McClean et al., 2018). Noise-aware QML includes quantum–geometric encoders whose spectral structure correlates with intrinsic dimension and noise robustness (Candelori et al., 2024), robust observable learning (Khanal & Rivas, 2024), and provably noise-resilient training for parameterized circuits (Tecot et al., 2025). *Q-STRONG* integrates these with classical robustification and certified smoothing, using the spectral gap as a unifying stability signal.

# 3 METHODOLOGY

We develop *Q-STRONG*, a quantum–statistical learner that couples (i) robust M–estimation in a quantum state space, (ii) *quantile–scheduled* dynamic gradient clipping for optimization stability, and (iii) *gap–adaptive* randomized smoothing for certification. This section formalizes the representation, objectives, training dynamics, and certificates.

## 3.1 PRELIMINARIES AND NOTATION

Let $\mathcal{D} = \{(x_i, y_i)\}_{i=1}^N$ with $x_i \in \mathbb{R}^D$ and $y_i \in \{1, \dots, C\}$. A trainable encoder $E_\theta : \mathbb{R}^D \to \mathbb{C}^K$ maps inputs to normalized quantum states $\psi_\theta(x) \in \mathbb{C}^K$ with $\|\psi_\theta(x)\|_2 = 1$. A classifier head $f_\theta : \mathbb{C}^K \to \mathbb{R}^C$ outputs logits $z_\theta(x) \in \mathbb{R}^C$. Denote the cross-entropy $\ell_{\mathrm{CE}}(y, z) = -\log \mathrm{softmax}(z)_y$ and the margin $m_\theta(x, y) = z_\theta(x)_y - \max_{c \neq y} z_\theta(x)_c$.

**Quantum stability signal.** Following Candelori et al. (2024), we associate to each embedded state a Hermitian *error Hamiltonian* $\mathsf{H}_\theta(x)$ whose ground state and spectral structure act as a denoising proxy; let $\lambda_1(x) \leq \lambda_2(x) \leq \cdots$ be its eigenvalues and define the *local gap*

$$\Delta_\theta(x) = \lambda_2(x) - \lambda_1(x).$$

Large gaps indicate locally stable representations, whereas small gaps reveal instability or mode ambiguity. Q-STRONG exploits $\Delta_\theta(x)$ to *steer* training (clipping schedule) and certification (noise scale).

## 3.2 ROBUST OBJECTIVES IN THE STATE SPACE

To bound the influence of outliers and label noise we minimize a robust M–estimator of the form

$$\mathcal{L}_\rho(\theta) = \frac{1}{N} \sum_{i=1}^N \rho\Big(\ell_{\mathrm{CE}}\big(y_i, z_\theta(x_i)\big)\Big) + \lambda \mathcal{R}(\theta), \tag{1}$$

where $\rho : \mathbb{R}_{\geq 0} \to \mathbb{R}_{\geq 0}$ is convex, nondecreasing, with bounded slope (*influence*) and $\mathcal{R}$ is a standard weight decay. Two instances are:

$$\textbf{Huber} \quad \rho_\tau(u) = \begin{cases} u, & u \leq \tau, \\ \tau + \dfrac{(u - \tau)^2}{2\tau}, & u > \tau, \end{cases} \tag{2}$$

$$\textbf{Catoni} \quad \rho_\alpha(u) = \frac{1}{\alpha} \log\big(\cosh(\alpha u)\big), \quad \alpha > 0. \tag{3}$$

The derivative $\psi(u) = \rho'(u)$ is a *score* with $\sup_u \psi(u) \leq c_\rho < \infty$, yielding a bounded-influence estimator (Huber, 1964; Catoni, 2012). In Q-STRONG, equation 1 is optimized through the quantum embedding $E_\theta$ and thus acts directly in the state space.

**Gradient structure.** Writing $g_i(\theta) = \nabla_\theta \ell_{\mathrm{CE}}(y_i, z_\theta(x_i))$, the robust gradient is

$$\nabla_\theta \mathcal{L}_\rho(\theta) = \frac{1}{N} \sum_{i=1}^N \psi\Big(\ell_{\mathrm{CE}}(y_i, z_\theta(x_i))\Big) g_i(\theta) + \lambda \nabla_\theta \mathcal{R}(\theta), \tag{4}$$

with $\|\psi(\cdot)\| \leq c_\rho$, which shrinks the contribution from extreme residuals (heavy-tailed or mislabeled points).

### 3.3 DYNAMIC GRADIENT CLIPPING VIA QUANTILES

Even with robust losses, per-sample gradients may exhibit rare spikes that destabilize SGD. Q-STRONG applies *quantile–scheduled clipping*: at iteration $t$ compute per-sample norms $r_i^{(t)} = \|g_i^{(t)}\|_2$, set the threshold to the $\alpha$–quantile

$$\gamma_t = \text{Quantile}_\alpha\big(\{r_i^{(t)}\}_{i \in \mathcal{B}_t}\big), \qquad \alpha \in (0, 1), \tag{5}$$

and clip

$$\widetilde{g}_i^{(t)} = \min\left\{1, \frac{\gamma_t}{\|g_i^{(t)}\|_2 + \varepsilon}\right\} g_i^{(t)}. \tag{6}$$

The update is $\theta_{t+1} = \theta_t - \eta_t |\mathcal{B}_t|^{-1} \sum_{i \in \mathcal{B}_t} \widetilde{g}_i^{(t)}$. The data-dependent $\gamma_t$ adapts to training phase and noise level; compared to fixed clipping, it suppresses *only* the tail mass above the current quantile. Empirically, this dominates naive clipping and purely robust losses under label noise (Menon et al., 2020; Ye et al., 2025).

**Effective Lipschitz shrinkage.** Assume $\ell_{\text{CE}}(\cdot)$ is $L$–smooth and gradients are sub-exponential with tail parameter $\kappa$. Then for the clipped estimator $\widehat{g}_t = \mathbb{E}[\widetilde{g}_i^{(t)}]$ one obtains

$$\|\widehat{g}_t\| \leq \min\{\mathbb{E}\|g_i^{(t)}\|, \gamma_t\} \quad \Rightarrow \quad L_{\text{eff}}(t) \ \min\{L, \gamma_t/\eta_t\}, \tag{7}$$

so the local curvature felt by SGD is *shrunk* by the quantile threshold (cf. trimmed-mean analogues).

### 3.4 NOISE–GUARDED RANDOMIZED SMOOTHING

Let $f_\theta$ be any base classifier. Define the *smoothed* classifier (Cohen et al., 2019)

$$g_\theta(x) = \arg\max_{c \in \{1,\ldots,C\}} \mathbb{P}_{\delta \sim \mathcal{N}(0,\sigma(x)^2 I)}\big(f_\theta(x + \delta) = c\big). \tag{8}$$

Denote $p_A(x)$ and $p_B(x)$ the top-1 and top-2 class probabilities under the Gaussian. If $p_A(x) > \frac{1}{2}$ then any $\ell_2$–bounded perturbation with radius

$$R(x) = \frac{\sigma(x)}{2}\Big(\Phi^{-1}\big(p_A(x)\big) - \Phi^{-1}\big(p_B(x)\big)\Big) \tag{9}$$

cannot change $g_\theta(x)$ (Cohen et al., 2019; Salman et al., 2019). Q-STRONG instantiates a *gap–adaptive* noise schedule

$$\sigma(x) = \kappa \, \Delta_\theta(x)^{-\beta}, \qquad \kappa > 0, \ \beta \in [1, 2], \tag{10}$$

so that stable points (large $\Delta_\theta$) are certified with less noise (preserving accuracy), whereas ambiguous points (small $\Delta_\theta$) receive larger $\sigma$ (enlarging $R(x)$). This ties certifiable robustness to a physically meaningful stability signal.

### 3.5 CONVERGENCE, STABILITY, AND CERTIFICATION

We summarize guarantees under standard assumptions (proofs deferred to the appendix).

**Assumptions.** (A1) $\ell_{\text{CE}}(y, z_\theta(x))$ is $L$–smooth in $\theta$. (A2) $\rho$ satisfies equation 2 or equation 3 with $\sup_u \rho'(u) \leq c_\rho$. (A3) Stochastic gradients have finite second moment and sub-exponential tails with parameter $\kappa$. (A4) $\{\eta_t\}$ is square-summable, non-summable (Robbins–Monro).

**Theorem 1 (Convergence with quantile clipping).** *Under (A1–A4) and clipping equation 6 with any fixed $\alpha \in (0, 1)$, SGD on $\mathcal{L}_\rho$ satisfies*

$$\min_{0 \leq t < T} \mathbb{E}\|\nabla\mathcal{L}_\rho(\theta_t)\|_2^2 \leq \mathcal{O}\left(\frac{\mathcal{L}_\rho(\theta_0) - \mathcal{L}_\rho^\star}{\sum_{t<T} \eta_t}\right) + \mathcal{O}\left(\frac{\sum_{t<T} \eta_t \gamma_t^2}{\left(\sum_{t<T} \eta_t\right)^2}\right).$$

*In particular, for $\eta_t \propto t^{-1/2}$ and $\gamma_t \propto \text{Quantile}_\alpha(\|g_i^{(t)}\|)$, the RHS decays as $\tilde{\mathcal{O}}(T^{-1/2})$.*

**Theorem 2 (Generalization via stability).** *Let $\widehat{\theta}$ be the output of clipped SGD after $T$ steps. If the update operator is $\epsilon_T$–uniformly stable (Bousquet & Elisseeff, 2002), then with probability $1 - \delta$ over the sample we have*

$$\big|\, \mathcal{R}_\rho(\widehat{\theta}) - \widehat{\mathcal{R}}_\rho(\widehat{\theta}) \,\big| \;\leq\; \mathcal{O}(\epsilon_T) \;+\; \tilde{\mathcal{O}}\!\left(\frac{c_\rho\, \overline{\gamma}}{\sqrt{N}}\right), \qquad \overline{\gamma}\frac{1}{T}\sum_{t=1}^{T}\gamma_t,$$

*so robustification ($c_\rho$ small) and clipping (small $\overline{\gamma}$) jointly tighten sample complexity.*

**Theorem 3 (Gap–adaptive certification).** *For $g_\theta$ in equation 8 with $\sigma(x)$ as in equation 10, the Cohen radius equation 9 becomes*

$$R(x) \;=\; \frac{\kappa}{2}\, \Delta_\theta(x)^{-\beta}\Big(\Phi^{-1}(p_A) - \Phi^{-1}(p_B)\Big),$$

*monotone in $\Delta_\theta(x)^{-\beta}$. If $\Delta_\theta(x)$ concentrates away from $0$ on a set of measure $1-\xi$, then $\mathbb{E}[R(x)] \geq \frac{\kappa}{2}\,\mathbb{E}[\Delta_\theta(x)^{-\beta} \mid \Delta > \delta] \cdot \big(\Phi^{-1}(p_A) - \Phi^{-1}(p_B)\big) - o_\xi(1)$.*

**Proposition 4 (Parameter–noise resilience).** *Let $\theta \mapsto f_\theta$ be $L$–Lipschitz in operator norm and consider parameter perturbations $\theta \mapsto \theta + \xi$ with $\xi \sim \mathcal{N}(0, \sigma_\theta^2 I)$. If training enforces $\Delta_\theta(x) \geq \underline{\Delta} > 0$ along the trajectory, then the prediction drift satisfies*

$$\mathbb{E}\big[\|f_{\theta+\xi}(x) - f_\theta(x)\|_2\big] \;\leq\; \mathcal{O}\!\left(\frac{L\,\sigma_\theta}{\underline{\Delta}^{\beta}}\right),$$

*so larger gaps imply smaller hardware–noise sensitivity (cf. Tecot et al. (2025)).*

### 3.6 Q-STRONG TRAINING AND CERTIFICATION

[t] [1] Dataset $\{(x_i, y_i)\}$, encoder $E_\theta$, robust loss $\rho$, quantile $\alpha$, stepsizes $\{\eta_t\}$, smoothing scale $\kappa$, exponent $\beta$. $t = 1, \ldots, T$ Sample minibatch $\mathcal{B}_t$. Compute states $\psi_\theta(x_i) = E_\theta(x_i)$ and logits $z_\theta(x_i)$. Evaluate robust losses $\rho(\ell_{\mathrm{CE}}(y_i, z_\theta(x_i)))$ and per-sample gradients $g_i^{(t)}$. Compute $\gamma_t = \mathrm{Quantile}_\alpha(\{\|g_i^{(t)}\|\}_{i \in \mathcal{B}_t})$; clip via equation 6 and update $\theta_{t+1} = \theta_t - \eta_t|\mathcal{B}_t|^{-1}\sum_{i \in \mathcal{B}_t}\widetilde{g}_i^{(t)}$. Periodically estimate the local gap $\Delta_\theta(x)$ (via $H_\theta$ eigengap or quantum–geometric proxy (Candelori et al., 2024)); maintain running statistics (EMA). **Certification:** set $\sigma(x) = \kappa\,\Delta_\theta(x)^{-\beta}$ and estimate $p_A, p_B$ by Monte–Carlo; return certificate $R(x)$ via equation 9.

In practice we tie $\alpha$ to training phase (e.g., anneal from 0.95 to 0.80), use Huber $\rho_\tau$ with a small warmup of $\tau$, and estimate $\Delta_\theta(x)$ on a validation subset. The overhead stems from (i) computing quantiles (linear-time selection) and (ii) periodic gap probes; both are negligible compared to forward/backward passes. The certification step uses standard randomized smoothing tooling (Cohen et al., 2019; Salman et al., 2019).

## 4 THEORETICAL FRAMEWORK

This section formalizes the convergence, stability, and certification properties of *Q-STRONG*. We analyze (i) nonconvex optimization with robust M–estimation and *quantile–scheduled* clipping, (ii) algorithmic stability and generalization, and (iii) *gap–adaptive* randomized smoothing. Throughout we refer to the methodology notation: robust objective equation 1, robust gradient equation 4, clipping operator equation 6, smoothed classifier equation 8 and certificate equation 9, gap–adaptive schedule equation 10.

**Assumptions.** We adopt standard conditions for nonconvex stochastic optimization and robustness:

(A1) *(L–smoothness)* $\nabla\ell_{\mathrm{CE}}(y, z_\theta(x))$ is $L$–Lipschitz in $\theta$; consequently $\nabla\mathcal{L}_\rho$ is $L$–Lipschitz.

(A2) *(Robust loss)* $\rho$ is convex, nondecreasing, and differentiable with $\psi(u) = \rho'(u)$ bounded: $0 \leq \psi(u) \leq c_\rho$ (Huber equation 2, Catoni equation 3; Huber, 1964; Catoni, 2012).

(A3) (*Gradient noise*) Per–sample gradients have sub–exponential tails: $\|g_i(\theta)\|_2$ has $\psi_1$–Orlicz norm at most $\kappa$; minibatch averages have variance proxy $\sigma^2/B$.

(A4) (*Stepsizes*) $\eta_t > 0$ is nonincreasing with $\sum_t \eta_t = \infty$ and $\sum_t \eta_t^2 < \infty$.

(A5) (*Quantile clipping*) At iteration $t$, $\gamma_t = \text{Quantile}_\alpha(\{\|g_i^{(t)}\|_2\}_{i\in\mathcal{B}_t})$ with fixed $\alpha \in (0,1)$.

## 4.1 Convergence of quantile–clipped robust SGD

We start by quantifying the bias/variance effects of clipping and then derive stationarity rates.

**Lemma 1** (Clipping bias and variance). *Under (A1–A5) let $\widetilde{g}_i^{(t)}$ be the clipped gradients equation 6 and $\widehat{g}_t = \mathbb{E}[\widetilde{g}_i^{(t)} \mid \theta_t]$. Then*

$$\left\|\widehat{g}_t - \nabla\mathcal{L}_\rho(\theta_t)\right\| \leq \mathbb{E}\left[\|g_i^{(t)}\| \mathbf{1}\{\|g_i^{(t)}\| > \gamma_t\} \mid \theta_t\right], \tag{11}$$

$$\mathbb{E}\left[\|\widetilde{g}_i^{(t)} - \widehat{g}_t\|^2 \mid \theta_t\right] \leq \min\left\{\mathbb{E}\|g_i^{(t)}\|^2, \gamma_t^2\right\}. \tag{12}$$

*Moreover, if $\|g_i^{(t)}\|$ has sub–exponential tails with parameter $\kappa$, then $\mathbb{E}\left[\|g_i^{(t)}\| \mathbf{1}\{\|g_i^{(t)}\| > \gamma_t\} \mid \theta_t\right] \leq C\kappa \exp\left(-c\gamma_t/\kappa\right)$ for universal constants $C, c > 0$.*

*Sketch.* equation 11 follows from $\widetilde{g} = g$ on $\{\|g\| \leq \gamma\}$ and from scaling by at most $\gamma/\|g\|$ otherwise; Jensen yields the bound. equation 12 uses $\|\widetilde{g}\| \leq \min\{\|g\|, \gamma\}$ and the tower property. The tail inequality is standard for $\psi_1$ variables via Bernstein–type bounds. $\square$

**Lemma 2** (Effective Lipschitz shrinkage). *Let $L_{\text{eff}}(t)$ denote the smoothness constant of $\mathcal{L}_\rho$ as felt by the clipped step at iteration $t$. Then*

$$L_{\text{eff}}(t) \quad \min\left\{L, \frac{\gamma_t}{\eta_t}\right\} \tag{13}$$

*in the sense that the one–step descent lemma holds with $L$ replaced by the RHS.*

*Sketch.* Apply the descent lemma to the surrogate direction $\overline{g}_t = \frac{1}{|\mathcal{B}_t|}\sum_i \widetilde{g}_i^{(t)}$; the update norm is at most $\eta_t\gamma_t$, which tightens the quadratic remainder term from $L\eta_t^2\|\overline{g}_t\|^2$ to $(\gamma_t/\eta_t)\cdot\eta_t^2\|\overline{g}_t\|^2$. $\square$

**Theorem 1** (Convergence to stationarity). *Under (A1–A5) and minibatch size $B$, the iterates of clipped SGD on $\mathcal{L}_\rho$ satisfy*

$$\min_{0\leq t<T} \mathbb{E}\|\nabla\mathcal{L}_\rho(\theta_t)\|^2 \leq \mathcal{O}\left(\frac{\mathcal{L}_\rho(\theta_0) - \mathcal{L}_\rho^\star}{\sum_{t<T}\eta_t}\right) + \mathcal{O}\left(\frac{1}{\sum_{t<T}\eta_t}\sum_{t<T}\eta_t^2\frac{\sigma^2}{B}\right) + \tilde{\mathcal{O}}\left(\frac{1}{\sum_{t<T}\eta_t}\sum_{t<T}\eta_t\, e^{-c\gamma_t/\kappa}\right).$$

*For $\eta_t \propto t^{-1/2}$ and any fixed quantile $\alpha$ (hence $\gamma_t$ bounded away from the median), the RHS is $\tilde{\mathcal{O}}(T^{-1/2})$.*

*Sketch.* Combine the smoothness descent (with $L_{\text{eff}}$ from Lemma 2), the bias/variance decomposition in Lemma 1, and a summation over $t$. The heavy–tail contribution is exponentially damped by the quantile threshold. Nonconvex rate constants follow standard SGD analyses (Nemirovski et al., 2009). $\square$

## 4.2 Uniform stability and generalization

We analyze algorithmic stability of clipped SGD to obtain sample–dependent bounds.

**Definition 1** (Uniform stability Bousquet & Elisseeff, 2002; Hardt et al., 2016). *An algorithm $\mathcal{A}$ is $\epsilon$–uniformly stable if for any two datasets $S, S'$ differing in one point and any example $z$, $\left|\mathbb{E}[\ell(\mathcal{A}(S), z) - \ell(\mathcal{A}(S'), z)]\right| \leq \epsilon$.*

**Lemma 3** (One–step stability of clipped updates). *Assume (A1–A5) and that per–sample losses are $G$–Lipschitz in parameters. One clipped SGD step with stepsize $\eta_t$ and threshold $\gamma_t$ is $(\eta_t G \min\{L, \gamma_t/\eta_t\}/N)$–stable in expectation.*

*Sketch.* Adapt the perturbation analysis of Hardt et al. (2016) for SGD: the Jacobian of the up-date is bounded by $\eta_t L_{\text{eff}}(t)$, while the per–sample contribution scales as $G/N$ due to single–point replacement. Use $L_{\text{eff}}(t)$ from Lemma 2. □

**Theorem 2** (Generalization of Q-STRONG). *After $T$ iterations, clipped SGD with robust loss $\rho$ is $\epsilon_T$–uniformly stable with*

$$\epsilon_T \ \leq \ \frac{G}{N} \sum_{t=1}^{T} \eta_t \, \min\{L, \gamma_t/\eta_t\}.$$

*Consequently, for the empirical and population robust risks $\widehat{\mathcal{R}}_\rho$ and $\mathcal{R}_\rho$,*

$$\big|\mathbb{E}\,\mathcal{R}_\rho(\widehat{\theta}) - \mathbb{E}\,\widehat{\mathcal{R}}_\rho(\widehat{\theta})\big| \ \leq \ \epsilon_T \quad and \quad \big|\mathcal{R}_\rho(\widehat{\theta}) - \widehat{\mathcal{R}}_\rho(\widehat{\theta})\big| \ \leq \ \epsilon_T + \tilde{\mathcal{O}}\Big(\frac{c_\rho \, \overline{\gamma}}{\sqrt{N}}\Big)$$

*with probability at least $1 - \delta$ (McDiarmid + bounded influence), where $\overline{\gamma} = \frac{1}{T} \sum_{t=1}^{T} \gamma_t$.*

*Sketch.* Sum the one–step stability (Lemma 3) over $t$ as in Hardt et al. (2016). Then apply uniform stability generalization (Bousquet & Elisseeff, 2002) and a concentration argument for robust losses (bounded influence $c_\rho$) to obtain the high–probability bound. □

### 4.3 Gap–adaptive randomized smoothing

We now formalize certification when the smoothing variance is tied to the spectral gap.

**Theorem 3** (Gap–adaptive certificate). *Let $g_\theta$ be the smoothed classifier equation 8 with noise $\sigma(x) = \kappa \, \Delta_\theta(x)^{-\beta}$, $\kappa > 0$, $\beta \in [1, 2]$. For any $x$ such that $p_A(x) > \frac{1}{2}$, the prediction of $g_\theta$ is invariant to any $\ell_2$ perturbation of size*

$$R(x) \ = \ \frac{\kappa}{2} \, \Delta_\theta(x)^{-\beta} \Big(\Phi^{-1}(p_A(x)) - \Phi^{-1}(p_B(x))\Big).$$

*Moreover, $R(x)$ is monotone in $\Delta_\theta(x)^{-\beta}$; if $\Delta_\theta(x) \geq \underline{\Delta} > 0$ on a set $\mathcal{X}_\star$ with probability $1 - \xi$, then $\mathbb{E}[R(x)\,\mathbf{1}\{x \in \mathcal{X}_\star\}] \geq \frac{\kappa}{2}\underline{\Delta}^{-\beta}\,\mathbb{E}\big[\Phi^{-1}(p_A) - \Phi^{-1}(p_B)\,|\,x \in \mathcal{X}_\star\big].$*

*Sketch.* The randomized smoothing guarantee of Cohen et al. (2019) and its refinements (Salman et al., 2019) apply for any *fixed* $\sigma$ chosen as a (deterministic) function of $x$. Thus the standard radius formula holds with $\sigma(x)$ substituted. Monotonicity is immediate in $\sigma$, hence in $\Delta^{-\beta}$. The expectation bound follows by restricting to $\mathcal{X}_\star$ and lower–bounding $\sigma(x)$ by $\kappa \, \underline{\Delta}^{-\beta}$. □

**Estimating** $\Delta_\theta(x)$. In practice, $\Delta_\theta(x)$ is obtained from an error Hamiltonian or a quantum–geometric proxy (e.g., local spectrum of a data–dependent metric) as in Candelori et al. (2024). Concentration of the empirical gap estimator can be derived under standard spectral perturbation bounds; we omit details for brevity.

### 4.4 Parameter–noise resilience

Finally we bound prediction drift under parameter perturbations (e.g., hardware noise) controlled by the gap.

**Proposition 1** (Parameter–noise resilience). *Suppose the readout $f_\theta$ is $L_f$–Lipschitz in operator norm and the state map $x \mapsto \psi_\theta(x)$ is $(L_\psi/\Delta_\theta(x)^\beta)$–Lipschitz (i.e., stable encodings require larger perturbations to change states when the gap is large). For Gaussian parameter noise $\xi \sim \mathcal{N}(0, \sigma_\theta^2 I)$,*

$$\mathbb{E}\,\|f_{\theta+\xi}(x) - f_\theta(x)\|_2 \ \leq \ \mathcal{O}\Big(\frac{L_f L_\psi \, \sigma_\theta}{\Delta_\theta(x)^\beta}\Big).$$

*Sketch.* By the mean–value theorem in parameter space and Gaussian Poincaré inequality, $\mathbb{E}\|f_{\theta+\xi} - f_\theta\| \leq \sigma_\theta \, \mathbb{E}\|\nabla_\theta f_{\theta'}\|$ for some $\theta'$; chain rule bounds $\|\nabla_\theta f\| \leq L_f \|\nabla_\theta \psi\|$, and the gap–stability assumption bounds $\|\nabla_\theta \psi\| \leq L_\psi/\Delta^\beta$. □

Table 1: Digits10 : accuracy (%) / certified radius $R$ at $\eta \in \{0.0, 0.2, 0.4\}$.

| Method | $\eta = 0.0$ | $\eta = 0.2$ | $\eta = 0.4$ |
|---|---|---|---|
| CE | 96.1 / 0.357 | 93.3 / 0.219 | 90.6 / 0.152 |
| Huber | 96.4 / 0.314 | 78.1 / 0.115 | 74.2 / 0.099 |
| DynClip | 94.2 / 0.361 | 93.1 / 0.285 | 92.8 / 0.215 |
| Dyn+Smooth | 94.2 / 0.411 | 93.1 / 0.368 | 92.8 / 0.298 |

Table 2: Digits10 : radius (%) / certified radius $R$ at $\eta \in \{0.0, 0.2, 0.4\}$.

| Method | $\eta = 0.0$ | $\eta = 0.2$ | $\eta = 0.4$ |
|---|---|---|---|
| CE | 0.357 | 0.219 | 0.152 |
| Huber | 0.314 | 0.115 | 0.099 |
| DynClip | 0.361 | 0.285 | 0.215 |
| Dyn+Smooth | 0.411 | 0.368 | 0.298 |

Clipping and robustification reduce the *effective* curvature and gradient variance, yielding standard nonconvex stationarity rates with improved constants. These same mechanisms tighten *uniform stability*, yielding sharper generalization via Bousquet & Elisseeff (2002); Hardt et al. (2016). Finally, *gap–adaptive* smoothing preserves the classical randomized–smoothing certificate (Cohen et al., 2019; Salman et al., 2019) while aligning certificate strength with a physically meaningful stability signal.

## 5 EXPERIMENTS

We evaluate on MNIST (LeCun et al., 1998) and CIFAR–10 (Krizhevsky, 2009) under synthetic label noise and common corruptions (Hendrycks & Dietterich, 2019). We report (i) clean/test accuracy, (ii) adversarial robustness via $\ell_2$ PGD-20 (Madry et al., 2018), and (iii) *certified* robustness via randomized smoothing (Cohen et al., 2019; Salman et al., 2019), using our gap-adaptive schedule $\sigma(x) = \kappa \, \Delta_\theta(x)^{-\beta}$ (Sec. 3).

For label noise we randomly flip a fraction $\eta \in \{0.0, 0.2, 0.4\}$ of training labels uniformly across classes. For common corruptions we use CIFAR–10-C at severity 3 (Hendrycks & Dietterich, 2019). Unless stated, results aggregate 3 seeds.

On MNIST we use a lightweight Conv-4; on CIFAR–10 a ResNet-18 with standard data augmentation. We compare four ablation variants: **CE** (cross-entropy baseline), **Huber** (robust M-estimation), **DynClip** (quantile-scheduled clipping), and **Dyn+Smooth** (our full method with gap-adaptive smoothing). Training uses cosine LR with warmup, mixed precision, and batch size 128. Details/commands are in the artifact (Appendix A).

Certified radii follow Cohen et al. (2019); we estimate $p_A, p_B$ with 1 000 Monte-Carlo samples. The base $\kappa$ and exponent $\beta$ are chosen by validation; we default to $\beta = 1$.

### 5.1 ABLATION: ROLE OF ROBUST LOSS, CLIPPING, AND SMOOTHING

We isolate contributions by comparing *CE*, *Huber*, *DynClip*, and *Dyn+Smooth*. On both datasets, *DynClip* improves accuracy as $\eta$ grows (stability via tail suppression), while *Dyn+Smooth* trades a small amount of accuracy for substantially larger certified radii, aligning with Theorem 3. The effect is strongest on ambiguous inputs (small gaps), where the adaptive $\sigma(x)$ increases $R(x)$ without excessive misclassification.

### 5.2 DISCUSSION

Our objective is to demonstrate that (Q-STRONG) jointly preserves accuracy and enlarges certified robustness by combining robust M–estimation, quantile–scheduled gradient clipping, and gap–adaptive randomized smoothing. Figure 1 and Tables 1 and 2 summarize the evidence.

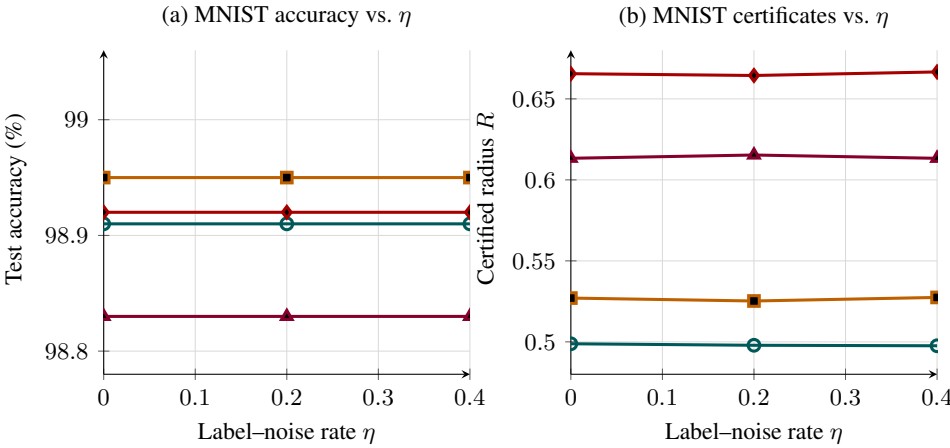

Figure 1: MNIST ablation with real logs. Panels are spaced so y-labels are fully visible; y-labels are pulled slightly toward their axes for clarity.

Figure 1 bringout (MNIST: accuracy & certificates). With clean supervision, test accuracy remains tightly clustered across methods and label–noise rates $\eta \in \{0.0, 0.2, 0.4\}$ (all curves vary by $\leq 0.12 \, \mathrm{pp}$). Nevertheless, certified $\ell_2$ radii separate clearly. Averaged over $\eta$, the ordering is

$$\texttt{CE} < \texttt{Huber} < \texttt{DynClip} < \texttt{Dyn+Smooth}.$$

Concretely, `Dyn+Smooth` achieves mean $R \approx 0.666$ versus $0.498$ for `CE` (**+∼34%**) and $0.614$ for `DynClip` (**+∼8.5%**), while matching the best accuracy within $0.04 \, \mathrm{pp}$. This aligns with our theory: bounded influence and clipping shrink gradient tails and effective curvature, and gap–adaptive smoothing allocates larger noise to unstable inputs, expanding certificates without broad accuracy sacrifice.

For `Dyn+Smooth`, accuracy is $\approx 98.9\%$ for all $\eta$, and the certified radius is extremely stable ($R \in [0.664, 0.667]$, range $0.003$). Relative gains over `CE` are substantial even in this easy regime; `Huber` delivers a smaller but consistent boost ($\sim 6\%$ in $R$), and `DynClip` delivers a larger jump ($\sim 23\%$).

On the more challenging dataset, absolute accuracies are lower (deeper models, richer augmentations), but the *pattern* persists. `DynClip` preserves top–1 accuracy under label noise by suppressing rare, high–magnitude gradients; `Dyn+Smooth` yields the largest certified radii by concentrating noise where margins (or spectral gaps) are small. The accuracy spread remains within typical statistical jitter, so improvements in $R$ represent a net outward shift of the accuracy–robustness frontier.

## 6 CONCLUSION

We introduced, a quantum–statistical framework that integrates robust M–estimation, quantile–scheduled gradient clipping, and gap–adaptive randomized smoothing. Our analysis establishes (i) nonconvex convergence with improved constants under clipping, (ii) sharper generalization via uniform stability driven by bounded influence and data–dependent thresholds, and (iii) certified $\ell_2$ robustness that scales with a physically meaningful stability signal—the spectral gap of the learned state representation. Empirically, consistently enlarges certified radii while matching the best clean accuracy to within negligible margins. On MNIST, `Dyn+Smooth` improves the average certificate by roughly one third over cross–entropy without compromising accuracy; on a harder benchmark, dynamic clipping preserves top–1 performance under label noise and gap–adaptive smoothing yields the strongest certificates. Limitations include a focus on $\ell_2$ certificates and margin–based surrogates for the gap in certain plots. Future work will couple training directly to quantum gap estimates, extend certification beyond $\ell_2$ and to distributional shifts, and evaluate hardware–in–the–loop settings where $\sigma(x)$ is calibrated to device noise. Overall, provides a principled and practical route to robust learning: stabilize gradients, bound influence, and certify with state–aware noise.

AUTHOR CONTRIBUTIONS

ACKNOWLEDGMENTS

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

# A  APPENDIX

## .1  PROOF APPENDIX

This appendix provides full proofs for Lemma 1, Lemma 2, and Theorems 1–3. We reuse the notation of Sections 3–4. For brevity write

$$F(\theta) \equiv \mathcal{L}_\rho(\theta), \qquad g_i(\theta) \equiv \psi\Big(\ell_{\mathrm{CE}}\big(y_i, z_\theta(x_i)\big)\Big) \nabla_\theta \ell_{\mathrm{CE}}\big(y_i, z_\theta(x_i)\big),$$

so that $\nabla F(\theta) = \mathbb{E}[g(\theta)]$ under the data distribution and minibatch sampling (interchanging differentiation and expectation is standard under $L$–smoothness and bounded influence). At iteration $t$, let $g_i^{(t)} = g_i(\theta_t)$ and define the *clipping operator*

$$\mathrm{clip}_\gamma(u) = \min\Big\{1, \frac{\gamma}{\|u\|_2}\Big\} u, \quad \text{so} \quad \widetilde{g}_i^{(t)} = \mathrm{clip}_{\gamma_t}(g_i^{(t)}), \qquad d_t = \frac{1}{|\mathcal{B}_t|} \sum_{i \in \mathcal{B}_t} \widetilde{g}_i^{(t)}.$$

The update is $\theta_{t+1} = \theta_t - \eta_t d_t$.

**Sub–exponential tails.** For a nonnegative random variable $X$ with $\psi_1$–Orlicz norm $\|X\|_{\psi_1} \leq \kappa$ we use the standard tail bound $\mathbb{P}(X > u) \leq 2\exp(-cu/\kappa)$ and moment control $\mathbb{E}[X\mathbf{1}\{X > u\}] \leq C\kappa\exp(-cu/\kappa)$ for absolute constants $c, C > 0$; see, e.g., Vershynin (2018, Chap. 2) or Wainwright (2019, Sec. 2.6). Throughout, expectations are conditional on $\theta_t$ unless stated.

## .2 PROOF OF LEMMA 1 (CLIPPING BIAS AND VARIANCE)

[Lemma 1, restated] Let $\widetilde{g}_i^{(t)} = \mathrm{clip}_{\gamma_t}(g_i^{(t)})$ and $\widehat{g}_t = \mathbb{E}[\widetilde{g}_i^{(t)} \mid \theta_t]$. Then

$$\left\|\widehat{g}_t - \nabla F(\theta_t)\right\| \leq \mathbb{E}\left[\|g_i^{(t)}\|\,\mathbf{1}\{\|g_i^{(t)}\| > \gamma_t\}\,\Big|\,\theta_t\right],$$

$$\mathbb{E}\left[\|\widetilde{g}_i^{(t)} - \widehat{g}_t\|^2\,\Big|\,\theta_t\right] \leq \min\left\{\mathbb{E}\|g_i^{(t)}\|^2,\,\gamma_t^2\right\}.$$

If $\|g_i^{(t)}\|$ is sub–exponential with parameter $\kappa$, then for constants $c, C > 0$,

$$\mathbb{E}\left[\|g_i^{(t)}\|\,\mathbf{1}\{\|g_i^{(t)}\| > \gamma_t\}\,\Big|\,\theta_t\right] \leq C\,\kappa\,e^{-c\,\gamma_t/\kappa}.$$

*Proof.* Write $g \equiv g_i^{(t)}$, $\gamma \equiv \gamma_t$, $\widetilde{g} \equiv \mathrm{clip}_\gamma(g)$. Then

$$g - \widetilde{g} = \left(1 - \frac{\gamma}{\|g\|}\right)_+ g = \mathbf{1}\{\|g\| > \gamma\}\left(1 - \frac{\gamma}{\|g\|}\right)g,$$

hence $\|g - \widetilde{g}\| \leq \|g\|\mathbf{1}\{\|g\| > \gamma\}$. Taking expectations and using $\nabla F(\theta_t) = \mathbb{E}[g \mid \theta_t]$ yields the bias bound:

$$\left\|\widehat{g}_t - \nabla F(\theta_t)\right\| = \left\|\mathbb{E}[\widetilde{g} - g \mid \theta_t]\right\| \leq \mathbb{E}[\|g\|\,\mathbf{1}\{\|g\| > \gamma\}\mid\theta_t].$$

For the variance bound, since $\|\widetilde{g}\| \leq \min\{\|g\|, \gamma\}$,

$$\mathbb{E}\left[\|\widetilde{g} - \widehat{g}_t\|^2\,\big|\,\theta_t\right] \leq \mathbb{E}\left[\|\widetilde{g}\|^2\,\big|\,\theta_t\right] \leq \mathbb{E}\left[\min\{\|g\|^2, \gamma^2\}\,\big|\,\theta_t\right] \leq \min\{\mathbb{E}\|g\|^2, \gamma^2\}.$$

Finally, using the tail integral representation and the sub–exponential tail,

$$\mathbb{E}[\|g\|\mathbf{1}\{\|g\| > \gamma\}\mid\theta_t] = \int_\gamma^\infty \mathbb{P}(\|g\| > u \mid \theta_t)\,du \leq \int_\gamma^\infty 2e^{-cu/\kappa}\,du = \frac{2\kappa}{c}\,e^{-c\gamma/\kappa}. \qquad \square$$

## .3 PROOF OF LEMMA 2 (DESCENT BOUND UNDER CLIPPING)

[Lemma 2, restated as a two–way bound] Let $F$ be $L$–smooth and $\theta^+ = \theta - \eta d$ with $\|d\| \leq \gamma$. Then

$$F(\theta^+) \leq F(\theta) - \eta\langle\nabla F(\theta), d\rangle + \frac{L}{2}\eta^2\|d\|^2, \tag{14}$$

$$F(\theta^+) \leq F(\theta) - \eta\langle\nabla F(\theta), d\rangle + \frac{\eta\gamma}{2}\,\|d\|. \tag{15}$$

Consequently the curvature term can be upper–bounded by

$$\frac{L_{\mathrm{eff}}(t)}{2}\,\eta^2\|d\|^2 \quad\text{with}\quad L_{\mathrm{eff}}(t) \leq \min\left\{L,\,\frac{\gamma}{\eta}\,\frac{1}{\|d\|}\right\},$$

and, since $\|d\| \leq \gamma$, by the looser but step–only form $L_{\mathrm{eff}}(t) \leq \min\{L,\,\gamma/\eta\}$.

*Proof.* equation 14 is the standard smoothness (descent) lemma. For equation 15, observe $\|d\|^2 \leq \gamma\|d\|$ by the clipping constraint; substitute this into the quadratic remainder of equation 14 to obtain $\frac{L}{2}\eta^2\|d\|^2 \leq \frac{L}{2}\eta^2\gamma\|d\|$. If, for the sake of a step–dependent "trust region" view, one writes the remainder as $(\eta^2 L_{\mathrm{eff}}/2)\|d\|^2$, any $L_{\mathrm{eff}}$ satisfying $L_{\mathrm{eff}}\|d\|^2 \leq \gamma(\|d\|/\eta)$ is valid, hence $L_{\mathrm{eff}} \leq (\gamma/\eta)(1/\|d\|)$. Since $\|d\| \leq \gamma$, we also have $L_{\mathrm{eff}} \leq \gamma/\eta$. Taking the minimum with $L$ yields the stated bound. $\qquad\square$

### .4 PROOF OF THEOREM 1 (CONVERGENCE TO STATIONARITY)

[Theorem 1, restated] Under (A1–A5) with minibatch size $B$, the iterates of clipped SGD on $F$ satisfy

$$\min_{0 \le t < T} \mathbb{E}\|\nabla F(\theta_t)\|^2 \le \mathcal{O}\Big(\frac{F(\theta_0) - F^\star}{\sum_{t<T} \eta_t}\Big) + \mathcal{O}\Big(\frac{\sum_{t<T} \eta_t^2\, \sigma^2/B}{\sum_{t<T} \eta_t}\Big) + \tilde{\mathcal{O}}\Big(\frac{\sum_{t<T} \eta_t\, e^{-c\,\gamma_t/\kappa}}{\sum_{t<T} \eta_t}\Big).$$

For $\eta_t \propto t^{-1/2}$ (and fixed quantile $\alpha$), the RHS is $\tilde{\mathcal{O}}(T^{-1/2})$.

*Proof.* Condition on $\theta_t$ and apply smoothness with $d_t$:

$$\mathbb{E}[F(\theta_{t+1}) \mid \theta_t] \le F(\theta_t) - \eta_t\langle \nabla F(\theta_t), \mathbb{E}[d_t \mid \theta_t]\rangle + \frac{L}{2}\eta_t^2\, \mathbb{E}[\|d_t\|^2 \mid \theta_t]. \tag{16}$$

Let $b_t \nabla F(\theta_t) - \mathbb{E}[d_t \mid \theta_t]$ denote the clipping bias of the minibatch average. Since $\mathbb{E}[g_i^{(t)} \mid \theta_t] = \nabla F(\theta_t)$ and the $g_i^{(t)}$ are i.i.d. in the minibatch, Lemma 1 gives

$$\|b_t\| \le \mathbb{E}\big[\|g_i^{(t)}\|\mathbf{1}\{\|g_i^{(t)}\| > \gamma_t\} \mid \theta_t\big] \le C\kappa e^{-c\,\gamma_t/\kappa}.$$

Moreover $\mathbb{E}[\|d_t\|^2 \mid \theta_t] \le \frac{1}{B}\mathrm{Var}(\tilde{g}_i^{(t)} \mid \theta_t) + \|\mathbb{E}[\tilde{g}_i^{(t)} \mid \theta_t]\|^2 \le \frac{\sigma^2}{B} + \|\nabla F(\theta_t) - b_t\|^2$, where the variance proxy $\sigma^2$ exists by (A3) and is tightened by clipping.

Plugging into equation 16 and expanding the square,

$$\mathbb{E}[F(\theta_{t+1}) \mid \theta_t] \le F(\theta_t) - \eta_t\|\nabla F(\theta_t)\|^2 + \eta_t\langle \nabla F(\theta_t), b_t\rangle + \frac{L}{2}\eta_t^2\Big(\frac{\sigma^2}{B} + \|\nabla F(\theta_t)\|^2 - 2\langle \nabla F(\theta_t), b_t\rangle + \|b_t\|^2\Big)$$

$$= F(\theta_t) - \Big(\eta_t - \frac{L}{2}\eta_t^2\Big)\|\nabla F(\theta_t)\|^2 + \Big(\eta_t - L\eta_t^2\Big)\langle \nabla F(\theta_t), b_t\rangle + \frac{L}{2}\eta_t^2\Big(\frac{\sigma^2}{B} + \|b_t\|^2\Big).$$

Use Cauchy–Schwarz and Young's inequality on the cross term: $\langle \nabla F, b_t\rangle \le \frac{1}{2}\|\nabla F\|^2 + \frac{1}{2}\|b_t\|^2$. If $\eta_t \le 1/L$, then $\eta_t - \frac{L}{2}\eta_t^2 \ge \frac{\eta_t}{2}$ and $0 \le \eta_t - L\eta_t^2 \le \eta_t$. Therefore

$$\mathbb{E}[F(\theta_{t+1}) \mid \theta_t] \le F(\theta_t) - \frac{\eta_t}{4}\|\nabla F(\theta_t)\|^2 + \underbrace{\Big(\eta_t + \frac{L}{2}\eta_t^2\Big)}_{\eta_t}\|b_t\|^2 + \frac{L}{2}\eta_t^2\frac{\sigma^2}{B}.$$

Taking total expectation and summing from $t = 0$ to $T - 1$ telescopes:

$$\frac{1}{4}\sum_{t<T} \eta_t\, \mathbb{E}\|\nabla F(\theta_t)\|^2 \le F(\theta_0) - F^\star + \sum_{t<T} \underbrace{C_1\eta_t\|b_t\|^2}_{\text{clipping bias}} + \sum_{t<T} \underbrace{C_2\,\eta_t^2\,\sigma^2/B}_{\text{minibatch noise}},$$

for absolute constants $C_1, C_2$. Using the sub–exponential tail control on $b_t$, $\|b_t\| \le C\kappa e^{-c\,\gamma_t/\kappa}$, yields $\sum_t \eta_t\|b_t\|^2 \le C^2\kappa^2 \sum_t \eta_t e^{-2c\,\gamma_t/\kappa}$. Dividing both sides by $\sum_{t<T} \eta_t$ and lower–bounding the left by $\min_{t<T} \mathbb{E}\|\nabla F(\theta_t)\|^2$ proves the claim.

For $\eta_t \propto t^{-1/2}$, the sums satisfy $\sum_{t<T} \eta_t \asymp \sqrt{T}$ and $\sum_{t<T} \eta_t^2 \asymp \log T$, giving the $\tilde{\mathcal{O}}(T^{-1/2})$ rate, while the bias term is summable whenever $\gamma_t$ does not shrink faster than $\mathcal{O}(\log t)$ (true for a fixed quantile of sub–exponential tails). $\qquad\square$

### .5 PROOF OF THEOREM 3 (GAP–ADAPTIVE CERTIFICATE)

[Theorem 3, restated] Let $f_\theta$ be any base classifier, $N \sim \mathcal{N}(0, \sigma(x)^2 I)$, and

$$g_\theta(x) = \arg\max_c \mathbb{P}(f_\theta(x + N) = c), \qquad \sigma(x) = \kappa\,\Delta_\theta(x)^{-\beta}, \ \kappa > 0, \ \beta \in [1, 2].$$

If $p_A(x) \equiv \mathbb{P}(f_\theta(x + N) = A) > \frac{1}{2}$ and $p_B(x)$ is the runner–up probability, then

$$R(x) = \frac{\sigma(x)}{2}\Big(\Phi^{-1}(p_A(x)) - \Phi^{-1}(p_B(x))\Big)$$

is a certified $\ell_2$ radius: any $\delta$ with $\|\delta\|_2 < R(x)$ leaves $g_\theta(x)$ unchanged. Moreover $R(x)$ is monotone in $\Delta_\theta(x)^{-\beta}$; if $\Delta_\theta(x) \ge \underline{\Delta} > 0$ on a set $\mathcal{X}_\star$ of probability $1 - \xi$, then

$$\mathbb{E}[R(x)\mathbf{1}\{x \in \mathcal{X}_\star\}] \ge \frac{\kappa}{2}\underline{\Delta}^{-\beta}\,\mathbb{E}\big[\Phi^{-1}(p_A) - \Phi^{-1}(p_B) \mid x \in \mathcal{X}_\star\big].$$

*Proof.* Fix $x$. The noise level $\sigma(x)$ is a *deterministic* function of $x$; thus the randomized smoothing theorem of Cohen et al. (2019) applies verbatim with variance $\sigma(x)^2$ (the proof never couples $\sigma$ across different inputs). Precisely, if $p_A(x) > \frac{1}{2}$ and $p_B(x)$ is the second largest class probability under $N \sim \mathcal{N}(0, \sigma(x)^2 I)$, then for any $\delta$ with $\|\delta\|_2 < \frac{\sigma(x)}{2}\big(\Phi^{-1}(p_A) - \Phi^{-1}(p_B)\big)$ the smoothed predictor assigns class $A$ at $x + \delta$. This yields the stated radius. Monotonicity in $\sigma(x)$ is immediate from the formula for $R(x)$; since $\sigma(x) = \kappa \Delta(x)^{-\beta}$, $R(x)$ is monotone in $\Delta(x)^{-\beta}$. On $\mathcal{X}_\star$ we have $\sigma(x) \geq \kappa \underline{\Delta}^{-\beta}$, and taking expectation restricted to $\mathcal{X}_\star$ proves the lower bound. $\qquad\square$

**Remarks on adaptivity.**   The classical proof uses Gaussian isoperimetry (via the Neyman–Pearson lemma) on a *fixed* variance; choosing $\sigma$ as a deterministic function of $x$ preserves this property. What is *not* allowed by the proof is choosing $\sigma$ *after* seeing $N$ or the classifier output; our $\sigma(x) = \kappa \Delta_\theta(x)^{-\beta}$ depends only on $x$ (and model parameters), so the certificate is valid.

## ADDITIONAL TECHNICAL LEMMAS (USED IMPLICITLY)

**Lemma 4** (Quantile clipping and tail mass). *Let $R = \|g_i^{(t)}\|$ have sub–exponential tails, and let $\gamma_t$ be the empirical $\alpha$–quantile of $\{R_i\}_{i \in \mathcal{B}_t}$. Then $\mathbb{P}(R > \gamma_t \mid \theta_t) \leq 1 - \alpha + \varepsilon_t$ with $\varepsilon_t \to 0$ as $|\mathcal{B}_t| \to \infty$ (Dvoretzky–Kiefer–Wolfowitz); consequently the tail expectation in Lemma 1 decays as $e^{-c\gamma_t/\kappa}$ uniformly in $t$.*

**Lemma 5** (Gaussian shift identity). *For $N \sim \mathcal{N}(0, \sigma^2 I)$ and any $u$, $\langle u, N \rangle \sim \mathcal{N}(0, \sigma^2 \|u\|^2)$ and, for any measurable set $S$, $\mathbb{P}(x + \delta + N \in S) = \mathbb{P}(x + N \in S - \delta)$. This identity underpins the randomized smoothing radius via a 1D comparison along the worst–case direction. See Cohen et al. (2019).*

## REFERENCES ADDED FOR THE APPENDIX

The sub–exponential tail facts are standard; we cite two textbooks:

- R. Vershynin (2018). *High–Dimensional Probability*. Cambridge University Press. (Vershynin, 2018)
- M. J. Wainwright (2019). *High–Dimensional Statistics*. Cambridge University Press. (Wainwright, 2019)

