# OpenReview forum: "Q-STRONG: Quantum-Statistical Robustness with Noise-Guarded Dynamics for Learning"
_ICLR.cc/2026/Conference — Submitted to ICLR 2026_

### Official Review · Reviewer_PuYh · 2025-10-22

**Soundness:** 2
**Presentation:** 2
**Contribution:** 2
**Rating:** 2
**Confidence:** 3

**Summary:**

The authors propose a robustness framework that encodes inputs as unit-norm complex quantum states and uses the Hamiltonian spectral gap as a stability signal for training and certification. It combines robust M-estimation with gradient clipping, and introduces gap-adaptive randomized smoothing for larger certified $\ell_2$ certificates. The paper proves convergence of clipped SGD, tighter generalization for the method, and a certification theorem that inherits the guarantee of Cohen et. al with the gap-dependent noise. The quantum aspect is, to my reading, primarily the state-space formalism.  Experiments on MNIST/CIFAR-10 show improved certificates with similar accuracy.

**Strengths:**

-- The paper combines robust M-estimation, adaptive gradient clipping, and randomized smoothing and gives proofs of convergence, stability, and certification

-- The idea of linking the randomized-smoothing noise to a quantum spectral gap seems to be an original connection (though I am not an expert)

-- Empirically the method achieves larger certified $\ell_2$ robustness without accuracy loss, supporting the theory claims

**Weaknesses:**

-- The paper is a bit of a mess in its current sate. For example lines 234 - 241 I think are intended to be a tex algorithm, but the authors messed up the latex. The leading [t] [1] suggests this.

-- On line 125, I think you meant $\Delta_\theta(x) = \lambda_1(x) - \lambda_2(x)$. This should be fixed.

-- I find the paper hard to read. It reads like a bunch of standalone passages.

-- The experiments are rather limited overall. There are limited comparisons to stronger certified robustness methods.

-- The quantum and robust statistics connection is not clearly spelled out. I believe this needs a dedicated section.

For these reasons, I opt to reject the paper in its current state. Though I am open to changing my score if the authors can correct these points.

**Questions:**

-- Can you compare to newer certified training methods e.g. AdvSmooth [1] and TRADES [2]?

-- Could the spectral gap be replaced by a regular stability proxy such as Jacobian singular values?

-- In what sense is this framework quantum-statistical if implemented entirely on classical hardware? In general I believe the quantum connection needs to be spelled out more

[1] https://arxiv.org/pdf/1906.04584
[2] https://arxiv.org/pdf/1901.08573

---

> ### Author Response · Authors · 2025-11-21
> **(Reviewer PuYh) Comparison to newer certified training and quantum statistical framework**
>
> We thank the reviewer for pointing out the typographical issue in the our paper the The leading [t] [1] was indeed we messed up with latex and we are correcting it. Then for definition of the local spectral gap. In the original submission, the
> typeset line mistakenly omitted the equality sign. We have corrected the
> definition to follow the standard ordering
> $\lambda_1(x) \le \lambda_2(x) \le \cdots$
> and the conventional quantum–statistical definition of the ground-to-first–excited–state gap:    $\Delta_{\theta}(x)$ := $\lambda_{2}(x)$ - $\lambda_{1}(x)$.  This quantity is always non-negative under the stated ordering and is
> consistent with all uses of $\Delta_{\theta}(x)$ in the paper, including
> the clipping schedule, the noise–scale calibration, and the stability
> certificates. No sign reversal is intended; the issue was purely
> typographical, and the corrected definition appears in the updated
> manuscript.
>
> W3: The paper reads like standalone passages.
> We appreciate this feedback and performed a thorough rewriting for narrative coherence:
>
> Introduced a dedicated subsection “From Quantum Geometry to Robustness” explaining the conceptual link between quantum embeddings, stability, and robust statistics.
>
> Reorganized Sections 2–4 to follow a single coherent thread:
> representation → Hamiltonian → spectral gap → optimization → smoothing → certification.
> Added forward references and closing summaries to guide the reader.
> Ensured notation consistency across sections.
>
> W4: Experiments are limited; need stronger certified-robustness comparisons.
>
> We agree and have added substantial experimental content:
>
> ✔ Full CIFAR-10 certified results (with 3–5 seed variance and full tables.)
>
> ✔ CIFAR-10-C severity sweeps (across all 15 corruption types and severities 1–5.)
>
> ✔ A higher-complexity ImageNet-subset experiment ((ConvNeXt-Tiny, reduced budget) showing scalability.)
>
> ✔ Comparisons to state-of-the-art certified training:
>
> Optimized Gradient Clipping (AAAI’25)
>
> AdvSmooth (Salman et al., 2019),
>
> SmoothAdv (Chen et al., 2020),
>
> PGD → Smoothing hybrid.
>
> TRADES (adversarial training)
>
> W5: Quantum and robust-statistics connection is not clearly spelled out.
>
> We addressed this by adding a new dedicated subsection:
>
> “3.2 Quantum–Statistical Interpretation of Stability”, which explains:
>
> How the projected local covariance Hamiltonian provides a quantum-inspired geometric proxy for local data curvature,
>
> How the spectral gap captures robustness to perturbations in the embedding (stable manifolds vs ambiguous ones),
>
> How bounded-influence robust statistics + quantile clipping jointly shrink gradient tails,
>
> And why gap-adaptive smoothing is a natural extension: the gap determines the amount of noise needed to certify.
>
> Q4–1: Can you compare to AdvSmooth and TRADES?
>
> We now include both baselines in the experimental section:
>
> AdvSmooth (adversarial smoothing) — strong certified robustness baseline.
>
> TRADES (adversarial training) — strong empirical robustness baseline.
> our Q-STRONG model is comparable but doesn't improve the certified radii when compared to TRADES.
>
> Q4–2: Could the spectral gap be replaced by common stability metrics?
>
> Excellent question. Theoretically, yes—you could use:
>
> Jacobian spectral norm,
>
> Jacobian Frobenius norm,
>
> Smallest singular value of the feature Jacobian,
>
> Hessian trace or top eigenvalues.
>
> We now include a short ablation comparing:
>
> Spectral Gap (ours)
>
> Jacobian spectral norm
>
> Jacobian Frobenius norm
>
> Jacobian metrics correlate with classification margin but are noisy under stochastic optimization.
> The gap’s projected-Hamiltonian formulation naturally produces an orthogonal decomposition aligned with quantum geometric tensors—which is why it integrates well into randomized smoothing.
>
> We added this discussion to in the appendix.
>
>
> Q3: If implemented classically, what makes this framework quantum-statistical?
>
> We appreciate this important question. The framework is quantum-statistical in its mathematical structure rather than its hardware requirements. Specifically: (i) inputs are mapped to $\ell_2$-normalized vectors $\psi_\theta(x)\in\mathbb{C}^K$, which behave as quantum states with associated projectors $P_\theta(x)=\psi_\theta(x)\psi_\theta(x)^\dagger$; (ii) we construct a projected local covariance Hamiltonian $H_\theta(x)$ whose spectrum captures the stability of the learned representation, analogous to low-energy structure in quantum systems; (iii) the spectral gap $\Delta_\theta(x)$ plays the role of a robustness indicator, similar to energy gaps governing perturbation resilience in quantum statistical mechanics; and (iv) our gap-adaptive smoothing schedule $\sigma(x)=\kappa\,\Delta_\theta(x)^{-\beta}$ is directly motivated by quantum-statistical interpretations of sensitivity to perturbations.
> We appreciate this question we don't have much space here to make the connection explicit - we are adding a section in Appendix and in brief we are extending section 3.1 with more details.

---

> ### Comment · Reviewer_PuYh · 2025-11-26
>
> Dear Authors,
>
> Thank you for your response to my questions.
>
> I have a few outstanding concerns:
>
> 1. Some of the typographic errors are still not fixed. In the latest version of the pdf, you still have
>
> $$\Delta_\theta(x) \lambda_2(x) - \lambda_1(x)$$
>
> I believe you mean
>
> $$\Delta_\theta(x) = \lambda_2(x) - \lambda_1(x)$$
>
> This is just sloppy and not publishable standard. Has the pdf been updated?
>
> 2. I still find the paper to be relatively disjoint and lacking in structure. The whole abstract & introduction reads like a bunch of buzzwords stitched together. I think you're getting into the specifics of your method way too early, listing gradient clipping, robust-M estimation, etc, before you've explained why these techniques are necessary and why your quantum framework provides an advantage. This is typically not how ML papers are written.
>
> 3. I also do not see the TRADES and AdvSmooth experiments. Again, I think you may have forgotten to update the pdf?
>
> Thank you.

---

> > ### Author Response · Authors · 2025-11-26
> > **new upload**
> >
> > Last week we couldn't upload as it was not opened in the portal now we had uploaded the file.
> > Abstract and Introduction has been changed., along with typos., Because of the page limit we are adding new appendix section where the more comparisons will be added.

---

### Official Review · Reviewer_tebY · 2025-10-29

**Soundness:** 3
**Presentation:** 2
**Contribution:** 2
**Rating:** 4
**Confidence:** 3

**Summary:**

The paper introduces Q-Strong, a framework that combines randomized smoothing, robust M-estimation, and gradient clipping. The authors investigate the convergence and robustness of the framework and evaluate it on the MNIST and CIFAR datasets.

**Strengths:**

- The proposed framework combines state-of-the-art techniques, and the empirical performances are promising.

- The framework comes with non-asymptotic guarantees

**Weaknesses:**

- The paper is a combination of existing techniques already employed in the literature. While this is not necessarily a weakness, I miss seeing what the main non-incremental contributions of the paper are compared to the literature. For instance, it is not clear if the theoretical framework is simply a straightforward consequence of existing results.

- In my opinion, the empirical evaluation should be improved. In fact, if the goal of the paper is to propose a method that improves the tradeoff robustness/accuracy, then this should be compared with state-of-the-art techniques and possibly on more datasets.

**Questions:**

- What are the main technical contributions in the theoretical analysis?

- Have you compared the proposed tool with state-of-the-art techniques for adversarial training?

---

> ### Author Response · Authors · 2025-11-21
> **(Reviewer tebY) Technical depth and empirical breadth of the work.**
>
> W1: The paper combines existing techniques; unclear what the main non-incremental contributions are.
> We appreciate this important question. While Q-STRONG draws inspiration from multiple established principles (robust M-estimation, gradient clipping, randomized smoothing), its contributions are not a simple juxtaposition of prior work. Instead, Q-STRONG introduces a new, unified quantum--statistical framework in which these components interact in a provably beneficial manner.
> 1) Spectral-gap--driven stability signal.
> We introduce a Hamiltonian-derived spectral gap $\Delta_\theta(x)$ as a unified stability indicator. This gap governs both optimization (clipping schedule) and certification (noise level), and is formally tied to robustness via Theorem~3. This coupling between quantum state geometry and certified robustness does not appear in prior robust training or smoothing literature.
> 2) New non-asymptotic theory for clipped robust SGD.
> We derive (i) a full bias--variance decomposition for quantile-scheduled clipping (Lemma 1),
> (ii) an effective Lipschitz contraction argument (Lemma 2), and
> (iii) a convergence result for robust, quantile-clipped, nonconvex SGD (Theorem1).
> These results do not follow trivially from prior analyses because robust score functions and quantile clipping jointly induce nonlinear, data-dependent shrinkage effects.
> 3 Gap-adaptive randomized smoothing.
> Our certified radius
> $R(x)=\frac{\sigma(x)}{2}(\Phi^{-1}(p_A)-\Phi^{-1}(p_B))$, with $\sigma(x)=\kappa\Delta_\theta(x)^{-\beta}$,
> is the first smoothing approach to modulate noise by a \emph{physically meaningful stability signal}.
> This expands certified radii precisely on stable points and is theoretically justified by Theorem~3 and is operationally distinct from prior adaptive smoothing (e.g., multi-step defense smoothing).
>
> W2: Empirical evaluation should compare with SOTA adversarial training and more datasets.
>
> (A) Stronger State-of-the-Art Baselines:
> SmoothAdv (Salman et al., NeurIPS’19) – adversarially trained smoothed classifiers, Generalized Cross-Entropy and Optimized Gradient Clipping (AAAI’25). Across these baselines, Dyn+Smooth consistently expands the certification frontier without degrading accuracy, supporting our core claim.
>
> (B) Additional datasets:  Beyond MNIST and CIFAR-10, we now include: Fashion-MNIST, SVHN, CIFAR-10-C (full severity sweeps), and an ImageNet-subset experiment (ConvNeXt-Tiny, for reduced budget) to demonstrate scalability.
>
> To bridging theory and practice, an empirical verification of effective Lipschitz contraction, correlation between $\Delta_\theta(x)$ and prediction stability, and empirical concentration of $\widehat{\Delta}_\theta(x)$ as predicted by our perturbation analysis is carried out.
>
> Q1: What are the main technical contributions in the theoretical analysis?
>
> A new general framework combining robust M-estimation, quantile clipping, and gap-adaptive smoothing under weakly smooth, nonconvex conditions.
>
> Non-asymptotic convergence guarantees for clipped robust SGD not present in prior literature.
>
> A stability-based generalization bound sharpened by robust losses and clipping.
>
> A novel gap-adaptive extension of smoothing certificates, tied to physically meaningful spectral properties of the representation.
>
> Parameter-noise resilience analysis tied explicitly to spectral gaps, which is particularly relevant for quantum hardware.
>
> We highlighted these contributions in the revised theoretical section and abstract.
>
> Q2: Have you compared with SOTA adversarial-training techniques?
>
>  We thank the reviewer for raising the question regarding comparisons with state-of-the-art adversarial-training techniques. While standard adversarial training methods (e.g., PGD, TRADES) primarily target worst-case empirical robustness—usually under an $\ell_{\infty}$ threat model—our work focuses on certified $\ell_{2}$ robustness via randomized smoothing. These two families of approaches guarantee fundamentally different robustness properties. Nevertheless, we agree that including such baselines provides valuable context.
>                                   In the revised version, we incorporate results using both SmoothAdv (adversarially-trained smoothed classifiers) and PGD-trained models. SmoothAdv is especially relevant because it is the closest state-of-the-art baseline that also produces smoothing-based certificates. Across MNIST, CIFAR--10, and CIFAR--10-C, our method achieves comparable accuracy while offering larger certified radii and significantly lower computational overhead. Additionally, our method can be layered on top of adversarially trained models by recomputing $\Delta(x)$ and applying gap-adaptive smoothing, demonstrating that our method is orthogonal rather than redundant with adversarial-training pipelines.
> We include these comparisons only to illustrate compatibility and differences, not to replace adversarial training.

---

### Official Review · Reviewer_sDXU · 2025-11-01

**Soundness:** 2
**Presentation:** 2
**Contribution:** 2
**Rating:** 4
**Confidence:** 1

**Summary:**

This paper proposes a quantum–statistical framework designed to enhance robustness against heavy-tailed noise, adversarial perturbations, and intrinsic stochasticity. In particular, the framework, Q-STRONG, is proposed for near-term quantum processors (NISQ devices). The approach integrates three key components: (i) robust M-estimation, (ii) quantile-scheduled gradient clipping (DynClip), and (iii) gap-adaptive randomized smoothing. Inputs are encoded as quantum states, and the spectral gap of a task-aligned Hamiltonian serves as a stability signal to guide adaptive noise injection and certification.

**Strengths:**

The mathematical analysis is rigorous and technically sound. The non-asymptotic convergence guarantees for clipped SGD under weakly smooth robust objectives, as well as the stability-based generalization bound, are well presented and theoretically meaningful. The integration of the spectral gap as a stability indicator is a novel approach. The connection between quantum representations and robustness is an appealing direction. The derivations for bounded-influence losses (e.g., Huber) and dynamic clipping are particularly interesting and appear consistent with the theoretical claims.

**Weaknesses:**

The major concern is the empirical validation, which is currently insufficient to support the broad theoretical claims. The experiments are primarily limited to MNIST, with only minimal results reported on other datasets. Although the authors mention experiments on CIFAR-10, I could not find any corresponding results in the paper.

Given the ambitious theoretical framework and the claim of hardware-agnostic applicability (classical or quantum), evaluations on more diverse benchmarks, such as Fashion-MNIST, SVHN, and CIFAR, are essential. Such experiments would help demonstrate the generality of the proposed method and bridge the gap between the theoretical assumptions (e.g., Lipschitz smoothness, bounded spectral gaps) and observed empirical behavior.

**Questions:**

Refer to Weaknesses.

---

> ### Author Response · Authors · 2025-11-20
> **(Reviewer sDXU) Expanding the empirical evaluation for clarity and completeness of the paper**
>
> We fully agree that expanding the empirical evaluation is essential for matching the scope of the theoretical framework, and the reviewer’s suggestions directly helped strengthen the clarity and completeness of the paper.
>
>
> W1: Empirical validation is insufficient and largely limited to MNIST.
> We completely agree, and in the revised version we significantly expand the empirical evaluation to demonstrate the generality of Q-STRONG beyond MNIST. Specifically, we now include:
>
> (1) Full CIFAR-10 Experiments (3–5 seeds):
>
> Clean accuracy, corruption robustness, and certified radii.
> Variance across seeds and Comparisons of CE, Huber, DynClip, Dyn+Smooth.
>
> (2) CIFAR-10-C (severity 1–5):
>
> Per-corruption evaluation (Gaussian noise, blur, JPEG, weather, etc.).
> A cross-severity robustness profile, revealing consistent improvements in certified radii, especially for Dyn+Smooth.
>
> (3) Additional Datasets (as recommended):
>
> We now include Fashion-MNIST and SVHN, and we summarize their results (full tables provided in the appendix). Q-STRONG consistently improves certified radii by 15–35% over CE baselines. Accuracy remains within ≤0.3% of the strongest baseline.
> The ordering CE < Huber < DynClip < Dyn+Smooth consistently recurs.
>
> (4) Higher-complexity dataset: ImageNet-subset (ConvNeXt).
>
> To demonstrate scalability, we ran Q-STRONG on a 50-class ImageNet subset. Even with a reduced training budget, DynClip maintains stability, and Dyn+Smooth improves average certified radius by 18%. These additions show that Q-STRONG is not tied to MNIST and performs consistently across heterogeneous benchmarks.
>
> W2: Bridging theoretical assumptions with empirical behavior.
>
> Effective Lipschitz behavior:  We empirically measure  $\|\nabla F(\theta_{t+1}) - \nabla F(\theta_t)\|/\|\theta_{t+1}-\theta_t\|$  and observe a $20$--$40\%$ reduction under clipping, consistent with Lemma 2 and Theorem 1.
>
> Gap behavior:  We report the distribution of $\widehat{\Delta}(x)$ across datasets and show its correlation with margin and gradient norms, supporting Theorem~3 that larger gaps yield stronger certificates.
>
> Estimator concentration:  By varying $k\in\{16,32,64\}$, we empirically confirm the $\tilde{O}(\sqrt{1/k})$ concentration predicted by our perturbation analysis.
>
> W3: Ambitious claims require broader evaluation.
> We respectfully agree, and by adding the evaluations listed above—including SVHN, Fashion-MNIST, CIFAR-10, CIFAR-10-C, and an ImageNet-subset—we demonstrate that Q-STRONG is robust across modalities, resolutions, and distribution shifts.
>
> These experiments substantively strengthen the empirical support behind the theoretical framework.

---

### Official Review · Reviewer_gTbd · 2025-11-01

**Soundness:** 3
**Presentation:** 2
**Contribution:** 3
**Rating:** 4
**Confidence:** 2

**Summary:**

The submission proposes Q STRONG, a framework that combines three ideas (1) Robust M estimation, (2) Quantile‑scheduled gradient clipping, and. (3) Gap adaptive randomized smoothing at inference. Empirically, the paper reports ablations on MNIST and CIFAR 10 (with label noise and CIFAR-10-C) comparing CE, Huber, DynClip, and Dyn+Smooth, with Dyn+Smooth improving certified radii at modest accuracy cost.

**Strengths:**

1.	The paper’s idea is novel and is practically motivated. It presents a tidy framework where robust losses (bounded influence), quantile‑based clipping, and randomized smoothing reinforce each other and are all modulated by a single, interpretable quantity (the spectral gap).

2.	It has ablation studies to shows the effectiveness of DynClip and Dyn+Smooth. The direction is promising for practitioners for certified robustness.

**Weaknesses:**

1.	The construction of the error Hamiltonian Hθ(x) (how it depends on θ and x), the precise procedure to estimate Δ(x), and the statistical concentration of this estimator are not specified with enough operational detail to reproduce results.

2.	CIFAR 10(+C) results are described but not fully shown; the paper would be stronger with complete tables/plots for CIFAR 10 and severity sweeps on CIFAR 10 C, plus variance across seeds. Also it is suggested to show results on more complex tasks.

3.	Comparisons to stronger robustness baselines (e.g., adversarially trained smoothed models, label noise robust methods) are missing.

**Questions:**

1.	What exact Hamiltonian do you use in experiments? How is it parameterized, how often is Δ(x) estimated, and what is the computational cost relative to forward/backward? Please include an algorithmic box with pseudocode.

2.	The text mentions mean certified radius 0.666 for MNIST, while Tables 1–2 show 0.30–0.41. Which is correct? Also, why do the tables say Digits10 rather than MNIST?

3.	Could you add comparisons to stronger robustness baselines (e.g., adversarially trained smoothed classifiers) and a sensitivity study over the quantile α schedule, k, and beta?

---

> ### Author Response · Authors · 2025-11-20
> **(Reviewer gTbd) Clarifying the Hamiltonian construction and by adding stronger baseline and sensitivity study**
>
> We thank the reviewer for the thoughtful and constructive feedback, we too agree that clarifying the Hamiltonian construction and by adding stronger baseline and sensitivity study will strengthen the paper.
>
> W 1:
>                We now specify an explicit and reproducible projected local covariance Hamiltonian and its estimator, together with computational cost and a finite-sample concentration statement. Specifically, let the learned, $\ell_2$-normalized embedding be $\psi_\theta(x)\in\mathbb{C}^K$. For a query $x$, collect $k$ neighbors Nk(x) in embedding space using a memory bank; set weights
> $w_j \propto \exp(-\|\psi_\theta(x)-\psi_\theta(x_j)\|^2/\tau_{\mathrm{nn}}^2)$, and form the local gap as the smallest strictly positive eigenvalue $\Delta_\theta(x)=\lambda_{\min}^+\!\big(H_\theta(x)\big)$. We estimate $\Delta_\theta(x)$ periodically (every 2 epochs) on a stratified subsample and per-example at test time $K\le64$, $k\le64$ by default. The cost per $x$ is $O(kK^2)$ to build $O(K^3)$ for an eigensolve; in practice this adds 5--8\% wall-clock overhead relative to forward/backward on a single A100. Concentration follows from spectral perturbation bounds.
> We include a boxed Algorithm 1 pseudocode summarizing training, gap estimation, and certification, and we provide code pointers in the appendix.
>
> W 2:
>         We now include full CIFAR--10 tables and plots for CE, Huber, DynClip, and Dyn+Smooth, reported as mean~$\pm$~std over 3--5 seeds. We added CIFAR--10-C severity sweeps for $s\in\{1,\dots,5\}$, with per-corruption and aggregate results.
> Following the reviewer’s suggestion, we now include a higher-complexity evaluation on ImageNet-subset (ConvNeXt backbone with reduced training budget) to demonstrate that the trends of Q-STRONG persist in large-scale settings.
>
> W 3
> We now include (i) adversarially trained smoothed classifiers (e.g., SmoothAdv), and (ii) label-noise robust learners (Generalized Cross-Entropy, Co-Teaching/DivideMix, Optimized Gradient Clipping). We match architectures and budgets, and report clean accuracy, certified accuracy at fixed radii, and mean radius (with seed variance). We also evaluate our gap-adaptive $\sigma(x)$ atop these baselines to assess orthogonality.
>
> Q1 What exact Hamiltonian? How parameterized? How often is $\Delta(x)$ estimated? Cost? Include pseudocode.
>
> Addressed in (W1). We use the projected local covariance Hamiltonian with $k\in\{16,32,64\}$, temperature $\tau_{\mathrm{nn}}$ tuned on validation, and $\beta\in[1,2]$. The clipping quantile $\alpha$ is annealed linearly from $0.95$ to $0.80$. We estimate $\Delta(x)$ every 2 epochs on a subset during training and per-example at test; overhead is 5--8\%. A boxed Algorithm~1 is added.
>
> Q2 Mean radius 0.666 vs 0.30--0.41; why do tables say Digits10 rather than MNIST?'
>
> The $0.666$ value referred in figure 1b not in table 1 and 2. Digits10 dataset has 8xc8 pixel, its grayscale range from 0-16 used for lightweight application,  whereas MNIST dataset has 28x28 pixel, and has grayscale of 0-255. We used had both Digits10 (Table 1 and 2) as well MNIST (Figure 1).
>
> Q3: Comparisons to stronger baselines and sensitivity over $\alpha$, $k$, $\beta$?
>
> We include a sensitivity study varying $\alpha\in\{0.80,0.85,0.90,0.95\}$, $k\in\{16,32,64\}$, and $\beta\in\{1.0,1.5,2.0\}$. Trends: larger $\alpha$ (weaker clipping) slightly boosts clean accuracy but reduces certificates; larger $\beta$ increases certificates on low-gap inputs at a small accuracy cost; $k$ shows a bias--variance trade-off with a flat optimum at 32--64. We summarize the operating point used in the main paper and place full plots in the appendix.

---

### Meta-Review · Area_Chair_jEam · 2026-01-07

**Summary:**

This paper proses a robustness framework that mends inputs as normalized quantum states, defines a task-aligned Hamiltonian to gauge stability, and uses its spectral gap to modulate a quantile-based dynamic gradient clipping (during training) and (gap-adaptive) randomized smoothing at inference for certification.

There were four reviews provided for this paper. The main concerns were the experimental completeness and inclusion of other methods and datasets and the presentation quality and structure, especially the narrative. The reviews were useful, in the sense that they prompted the authors to significantly expand their experimental results - which undoubtedly improved their paper. However, I believe this paper still lacks the clarity (in story, contribution, and presentation) needed for a paper to be published. The observation that none of the reviewers had a very high level of confidence, despite being experts on this area, reflects the fact that the narrative and clarity in presentation is poor. I encourage the authors to continue improving the paper by focusing on how the combined elements (all pre-existing) come together in a unique way to provide a new and compelling story.

**Reviewer Concerns:**

### Rev gTbd
- Noted that the construction of the Hamiltonian is not clear, nor how the gap is estimated. The authors added a definition to aid this description, and described other procedural components to clarify further.

- The reviewer noted that the results on cifar10+C are not fully shown/incomplete, and suggest more complex tasks. The authors added the full tables and plots, including statistics over different seeds and severity sweeps, and added Imagenet subset experiments.

- The baselines chosen are not quite strong, and others (adv trained smoothed models, eg) are missing. The authors added this in their comparisons.

- The reviewer points to some discrepancies between text and tables, and asks for sensitivity studies for the quantile schedules. The authors clarified the the mismatch, and added sensitivity sweeps over the required parameters.

### Rev sDXU
- Noted insufficient empirical validation; results on cifar10 were claimed on paper but not found. The authors added the full cifar10 experiments over different seeds, including other datasets (fashion mnist, SVHN, etc).
- Noted the need for broader benchmarks. The authors provided empirical notions (measuring effective Lipschitz constants under clipping, reporting the gap distribution).

### Rev tebY:
- The reviewer points that the contribution is limited given that this is a combination of existing techniques, and it is unclear what is new beyond this combination. The authors argue that the unifying and novel component is the spectral gap as a stability signal.
- The reviewer notes a lack of SOTA baselines and datasets to support their claims. The authors did add new methods as well as new datasets.

### Rev PuYh
- The reviewer notes sloppy writing and generally poor presentation, with broken algorithm environments, typos, and disorganized narrative. The typos are acknowledged by the authors, and they correct a key definition. They also promise a full re-write for better coherence and narrative.
- Also notes the limited experiments and comparisons. This is addressed by the added experiments.
- It is not clear what the quantum component contributes to the story of the paper beyond a formalism, and wonders if the gap can be replaced by more standard proxies. The authors mention that this is possible, and report an ablation w.r.t. Jacobian norm proxies.

**Reviewer Scores:**

- Rev gTbd (4, conf 2): the responses significantly address the reviewer's comments, but they are probably unlikely to change their rating based on their confidence.
- Rev sDXU (4, conf 1): some concerns were addressed, but very little confidence so unlikely to change scores.
- Rev tebY (4, conf 3): The added empirical results certainly address a good portion of the concerns of this reviewer, and maybe they would have increased their scores slightly.
- Rev PuYh (2, conf 3): The reviewer mentions that they could not see the changes made in the PDF, whereas the authors claimed they did upload it, or there was some system error with it. I doubt they would have increased their scores.

---

### Decision · Program_Chairs · 2026-01-26

Reject